# Anti-Osteoarthritic Effects of Prunella Vulgaris and Gentiana Lutea In Vitro and In Vivo

**DOI:** 10.3390/antiox12010047

**Published:** 2022-12-26

**Authors:** Jeonghyun Kim, Chang-Gun Lee, Seokjin Hwang, Seung-Hee Yun, Laxmi Prasad Uprety, Kang-Il Oh, Shivani Singh, Jisu Yoo, Hyesoo Jeong, Yoonjoong Yong, Subin Yeo, Eunkuk Park, Seon-Yong Jeong

**Affiliations:** 1Department of Medical Genetics, Ajou University School of Medicine, Suwon 16499, Republic of Korea; 2Department of Biomedical Sciences, Ajou University Graduate School of Medicine, Suwon 16499, Republic of Korea; 3AI-Superconvergence KIURI Translational Research Center, Ajou University School of Medicine, Suwon 16499, Republic of Korea; 4Nine B Co., Ltd., Daejeon 34121, Republic of Korea

**Keywords:** osteoarthritis, anti-inflammatory effect, *Prunella vulgaris*, *Gentiana lutea*, destabilization of medial meniscus

## Abstract

Osteoarthritis (OA) is the progressive destruction of articular cartilage with severe symptoms, including pain and stiffness. We investigated the anti-osteoarthritic effects of *Prunella vulgaris* (PV) and Gentiana lutea (GL) extract in primary cultured chondrocytes RAW 264.7 cells in vitro and destabilization of the medial meniscus (DMM)-induced OA mice in vivo. Primary chondrocytes were induced with IL-1β, and RAW 264.7 cells were treated with LPS and co-incubated with either individual extracts of PV and GL or different ratios of PV and GL mixture. For the OA animal model, the medial meniscus (DMM) was destabilized in 9-week-old male C57BL/6 mice. Treatment of individual PV and GL and combination of PV and GL extracts inhibited the mRNA expression level of *COX2* in chondrocytes and RAW 264.7 cells. The optimized inhibitory effect was attained with a PV and GL combination at an 8:2 ratio (PG) without cytotoxic effects. PG extracts prevented the expression of catabolic factors (*COX2, Mmp3, Mmp9,* and *Mmp13*) and inflammatory mediator levels (PGE2 and collagenase). In addition, PG decreased subchondral sclerosis and increased BMD in the subchondral region of DMM-induced OA mice with protection of articular cartilage destruction by inhibiting inflammatory processes. This study suggests that PG may be an alternative medicinal herb for treatment of OA.

## 1. Introduction

Knee osteoarthritis (OA), also referred to as degenerative joint disease of the knee, is characterized by pain, stiffness, and changes in knee joint constructs, such as the synovium, ligament, and articular cartilage [1]. Approximately 37% people over 60 years of age have knee OA worldwide, and it is a global issue in an aging society [2]. Various risk factors for OA, such as heavy work or sports activities, injury, frequent kneeling, being overweight, and genetics, can stimulate mechanical stress on the knee joint, resulting in degeneration of the articular cartilage [3].

Joint instability caused by excessive physical kneeling promotes the formation of subchondral sclerosis and periarticular tissue abnormalities [4]. Damaged joint cells, such as chondrocytes, synovial fibroblasts, monocytes, and macrophages, induce the secretion of inflammatory cytokines such as interleukin-1-beta (IL-1β) [5]. In addition, IL-1β induced by external stress upregulates nuclear factor kappa B (NF-κB) and various mitogen-activated protein kinase (MAPK) signaling pathways, including extracellular signal-regulated kinase (ERK)**,** jun N-terminal kinase (JNK)**,** and p38 mitogen-activated protein kinases (p38^MAPK^), which can accelerate the inflammatory response [5]. Consistently, increased MAPKs have been observed in OA patients, and activated MAPKs are known to promote matrix metalloproteinase (Mmp) synthesis, which destroys articular cartilage [6].

Both IL-1β and NF-κB enhance the synthesis of OA-associated catabolic factors, such as cyclooxygenase-2 (COX2), matrix metalloproteinase 3 (Mmp3), matrix metalloproteinase 13 (Mmp13), nitric oxide (NO), and prostaglandin E2 (PGE2), by initiating the SRY-box transcription factor (Sox9) and the Src/PI3K/AKT pathway and inflammatory cytokines, including interferon gamma (IFN-γ), IL-1β, IL-6, and tumor necrosis factor-α (TNF-α) in chondrocytes [7,8,9]. These processes damage articular cartilage and accelerate OA [10]. 

A study revealed that OA treatment relieves inflammatory responses in OA pathogenesis [11]. For example, non-steroidal anti-inflammatory drugs (NSAIDs), such as cyclooxygenase inhibitors (dexamethasone, meloxicam, and aspirin), relieve pain and inhibit PGE2 secretion by ameliorating inflammatory responses in chondrocytes [12]. However, the long-term use of these drugs is limited because of their renal, cardiac, gastrointestinal, and hepatological side effects, leading to an increased risk of NSAIDs use [13].

Herbal plants have been widely used as alternative treatments because of their beneficial effects, such as relieving pain and decreasing inflammatory responses [14,15]. In addition, herbal plants have fewer side effects than chemical medications, supporting the increased demand for herbal plants [16]. Phytochemical studies have demonstrated that some single compounds derived from herbal plants, such as curcumin (*Curcuma longa*) and resveratrol (*Polygonum cuspidatum*), have been successfully researched for their anti-osteoarthritic effects, and the demand for the identification of new herbal plants for the prevention and treatment of OA has increased [17,18,19]. In addition, combined therapy with two or more plant extracts has shown synergistic effects on specific diseases, such as cardiovascular disease, cancer, and inflammation, to minimize the dosage of each extract [20,21,22].

*Prunella vulgaris* (PV) and *Gentiana lutea* (GL) are traditional medicines in east Asia. PV belongs to the Lamiaceae family, which is rich in polyphenols and promotes α-glucosidase inhibitory and antioxidant effects for treating type 2 diabetes [23,24]. A study reported that PV exerted anti-inflammatory properties in the RAW 264.7 cell inflammatory response by regulating PGE2 and NO secretion, which plays a central role in OA development [25]. GL, known as bitter root, yellow gentian, and bitterwort, belongs to the genus *Gentiana* in the family Gentianaceae [26]. Bioactive compounds of gentian promote anti-atherosclerotic [27], antioxidant [28], anti-obesity [29] and anti-inflammatory effects [27,28]. One of the main components isolated from GL showed protective effects against IL-1β-induced articular chondrocyte inflammation by reducing COX2, Mmp, and PGE2 expressions [30]. Although the anti-inflammatory effects of PV and GL on OA have been reported, the combined treatment of PV and GL on OA in vitro and in vivo has not been reported.

This study investigated the protective effects of PV and GL extracts on OA on catabolic factors and destructive mediators in primary cultured chondrocytes and RAW 264.7 cells in vitro. Subsequently, the inhibitory effects of PV and GL extracts on subchondral sclerosis, histological analysis, and inflammatory cytokines in a medial meniscus (DMM)-induced OA mouse model were examined in vivo.

## 2. Materials and Methods

### 2.1. Extraction and Confirmation of Standards Compounds from PV and GL Extract

PV extract powder, GL extract powder, and the combination (PG extract) were provided by NINE B Co., Ltd. (Daejeon, Republic of Korea). *PV* and *GL* were extracted with water and 30% ethanol, respectively. The extracts were sequentially processed using filtration, evaporated, and spray-dried with dextrin. The combined product, PG extract, was prepared by mixing PV and GL extract powder at a weight ratio (*w/w*) of 8:2. PV, GL, and PG extraction was performed by subjecting it to gradient water with a sequential 0.1% trifluoroacetic acid-acetonitrile solvent system (5. 20, 25, 40, 5, and 5% acetonitrile) using high-performance liquid chromatography (HPLC; Agilent Technologies, Santa Clara, CA, USA) with a Hypersil GOLD^TM^ column (Thermo Fisher Scientific, Waltham, MA, USA). The HPLC conditions were as follows: column temperature, 25 °C; detection wavelength, 280 nm, and flow rate, 0.6 mL/min. The standard compounds of rosmarinic acid for PV and Gentiopicroside for GL are shown in Appendix A, and the standard compounds of gentiopicroside (1) and rosmarinic acid (2) were compared with the combination of PV and GL with the chemical structures of gentiopicroside and rosmarinic acid (Appendix A).

### 2.2. Cell Culture and Primary Chondrocyte Isolation

RAW264.7 cells were obtained from the Korean Cell Line Bank (KCLB No. 46609) and maintained in cultured medium (high-glucose Dulbecco’s modified Eagle’s medium (Invitrogen, Carlsbad, CA, USA) containing 10% fetal bovine serum (Gibco) and 1% antibiotic-antimycotic (Invitrogen, Carlsbad, CA, USA)). Primary chondrocytes from the knee articular cartilage of 5 day-old mice were isolated using a culture medium containing 1% collagenase type II (Sigma-Aldrich, St. Louis, MO, USA) and 0.5% trypsin EDTA for 2 h at 37 °C, as previously described [31]. Digestive solutions were filtered using a Falcon^®^ 40 μm cell strainer (Corning, Riverfront Plaza Corning, NY, USA), and chondrocytes were obtained by centrifugation at 1200 rpm for 2 min. To induce an inflammatory response, chondrocytes were treated with IL-1β (1 ng/mL; Genscript, Piscataway, NJ, USA) for 2 d and RAW264.7 cells were stimulated with lipopolysaccharide (LPS, 1 μg/mL; Sigma-Aldrich, St. Louis, MO, USA) for 1 d.

### 2.3. Water-Soluble Tetrazolium Salt (WST) Assay

The cells were seeded in 96-well plates (1 × 10^5^ cells) at 37 °C in 5% CO_2_. After 3 d, the cells were incubated with three different concentrations of the PV and GL (PG) extract (50, 100, and 150 µg/mL) for 2 d. Cytotoxicity was measured using the D-Plus™ CCK cell viability assay kit (Dongin Biotech, Seoul, Korea) at an absorbance of 450 nm using a microplate reader (Bio-Rad, Hercules, CA, USA).

### 2.4. Reverse Transcription-Polymerase Chain Reaction (RT-PCR) and Quantitative Reverse-Transcription PCR (qRT-PCR)

Cells were treated with TRIzol reagent (Invitrogen, Carlsbad, CA, USA) to isolate total RNA, according to the manufacturer’s instructions. Complementary DNA (cDNA) was synthesized from 100 ng of RNA using a RevertAid™ H Minus First Strand cDNA synthesis kit (Fermentas, Hanover, NH, USA). Reverse transcriptase PCR (RT-PCR) was performed using HiPi Plus 5X PCR MasterMix (ELPIS Biotech, Daejeon, Republic of Korea), and quantitative RT-PCR was performed using the SYBR Green I qPCR kit (TaKaRa, Shiga, Japan). The specific primers used for this study were as follows: forward 5′-TCA CTG CCA CCC AGA-3′ and reverse 5′-TGT AGG CCA TGA GGT CCA C-3′ for mouse *Gapdh*, forward 5′-GGT CTG GTG CCT GGT CTG ATG AT-3′ and reverse 5′-GTC CTT TCA AGG AGA ATG GTG C-3ʹ for mouse *COX2*, forward 5′-CTG TGT GTG GTT GTG TGC TCA TCC TAC-3′ and reverse 5′-GGC AAA TCC GGT GTA TAA TTC ACA ATC-3′ for mouse *Mmp3*, forward 5′-CTT CGA CAC TGA CAA GAA GTG G-3′ and reverse 5′-GGC ACG CTG GAA TGA TCT AAG-3′ for mouse *Mmp9,* forward 5′-CTT CTT CTT GTT GAG CTG GAC TC-3′ and reverse 5′-CTG TGG AGG TCA CTG TAG ACT-3′ for mouse *Mmp13*. The mRNA expression levels were normalized to *Gapdh* expression, and the relative expression levels of the genes were calculated using the 2^−ΔΔCt^ method.

### 2.5. Western Blot Analysis

Cells were lysed with radioimmunoprecipitation buffer (RIPA; BIOSESANG, Seongnam, Republic of Korea), containing phenylmethylsulfonyl fluoride PMSF (Sigma-Aldrich, St. Louis, MO, USA) and a phosphatase inhibitor (Sigma-Aldrich, St. Louis, MO, USA). Total protein was separated using sodium dodecyl sulfate-polyacrylamide gel electrophoresis (SDS-PAGE) and transferred to polyvinylidene difluoride (PVDF) membranes. The membranes were immunoblotted with primary antibodies specific for COX2 (ab39012; Abcam, Waltham, MA, USA), NF-κB (#8242; Cell Signaling Technology, Danvers, MA, USA), p-NF-κB (#3033S; Cell Signaling Technology, Danvers, MA, USA), β-actin (sc-47778; Santa Cruz Biotechnology, Dallas, TX, USA), and appropriate secondary antibodies (horseradish peroxidase [HRP]-conjugated goat anti-mouse IgG (Bethyl Laboratories, Montgomery, TX, USA), and HRP-conjugated goat anti-rabbit IgG (Bethyl Laboratories, Montgomery, TX, USA)). Immunolabeled proteins were detected using West-Q Pico Dura ECL Solution (GenDEPOT, Katy, TX, USA).

### 2.6. Destabilization of Medial Meniscus (DMM) Mouse Model and Micro-CT Analysis

All animal experiments were approved by the Institutional Animal Care and Use Committee (IACUC) of Ajou University (2020-0036), and experiments were performed according to the guidelines of the committee. To induce osteoarthritis (OA) in a mouse model, the medial meniscus (DMM) was destabilized in 9-week-old male C57BL/6 mice. Mice were anesthetized with tiletamine/zolazepam (Zoletil; Vibrac Laboratories, Carros, France), and the medial meniscus of the left knee joint was dissected. After surgery, mice were fed food pellets containing three different concentrations of PG extract (50, 100, and 200 mg/kg/d) for 8 weeks. At the end of the experiment, the mice were sacrificed, plasma samples were stored at −80 °C, and the left knee joint was fixed with 4% paraformaldehyde (BIOSESANG) at 4 °C until analysis. For micro-CT analysis, articular knee joints were scanned using high-energy spiral scan micro-CT (Skyscan 1173; Bruker, Billerica, MA, USA) and representative CT images were reconstructed using CTvox software (version 3.2, Bruker, Billerica, MA, USA). The bone mineral density of the region of interest was measured using Skyscan 1173 software.

### 2.7. Histology

The formalin-fixed knee joint was decalcified using 0.5 M ethylenediaminetetraacetic acid (EDTA, pH 8.0) for 2 weeks, embedded in paraffin blocks, and sectioned (6 µm) using a rotary microtome (Leica Biosystems, Wetzler, Germany). Slide sections were subjected to safranin O staining to analyze the damaged regions of the articular cartilage. To determine the inflammatory responses of the articular cartilage, immunohistochemistry (IHC) was performed using specific primary antibodies, including COX2 (ab39012), Mmp3 (ab52915), and Mmp13 (ab39012) obtained from Abcam (Waltham, MA, USA). All stained slides were scanned using an Axio Scan 7 slide scanner (Carl Zeiss, Oberkochen, Germany). Cartilage destruction scoring was assessed using the Osteoarthritis Research Society International (OARSI) scoring to evaluate the degree of cartilage destruction [32].

### 2.8. NO and Plasma Cytokine Analysis

NO levels in the cellular supernatants were examined using a Nitrate/Nitrite Colorimetric Assay Kit (Cayman Chemical, Ann Arbor, MI, USA) according to the manufacturer’s instructions. Plasma levels of inflammatory cytokines (IFN-γ, IL-1-β, IL-6, and TNF-α) were analyzed using a mouse cytokine/chemokine magnetic bead panel immunology multiplex assay kit (MCYTOMAG-70k; Millipore, Billerica, MA, USA) and a MAGPIX^®^ multiplex analyzer (Luminex, Austin, TX, USA).

### 2.9. Statistical Analysis

All data in the bar graphs were expressed using GraphPad Prism 9.2.0 software (GraphPad Software, San Diego, CA, USA). Statistical significance was evaluated by one-way analysis of variance (ANOVA), followed by Tukey’s honestly significant difference (HSD) post hoc test, using professional Statistical Package software (SPSS 25.0 for Windows, SPSS Inc., Chicago, IL, USA).

## 3. Results

### 3.1. PV and GL Mixtures Inhibit COX2 mRNA Expression

Inflammation in mouse primary cultured chondrocytes is predominantly initiated by increased expression levels of IL-1β, and the inflammatory responses in RAW 264.7, macrophage have been induced by the administration of LPS [33,34]. First, we confirmed the effects of single extracts of PV and GL on inflammation in primary chondrocytes and RAW 264.7 cells and compared the synergistic effects of the combination of PV and GL extracts at three different ratios (6:4, 7:3, and 8:2). High amounts of PV were combined in the mixture extract because PV is a well-known traditional herbal medicine used for the treatment of inflammation [35]. Primary chondrocytes were induced with IL-1β (1 ng/mL), and RAW 264.7 cells were treated with LPS (1 µg/mL) and co-incubated with either individual extracts of PV and GL or different ratios of PV and GL mixture (100 and 150 µg/mL). Since cyclooxygenase-2 (COX2) is a reliable inflammatory biomarker that is detectable during the inflammatory response [36], we examined the anti-inflammatory effects of PV and GL extracts by evaluating the mRNA expression level of *COX2* in chondrocytes and RAW 264.7 cells using quantitative reverse transcription-polymerase chain reaction (qRT-PCR). As expected, individual PV and GL extracts inhibited *COX2* mRNA expression in both IL-1β-induced chondrocytes and LPS-induced RAW 264.7 cells (Appendix A). Additionally, combined PG extracts at ratios of 6:4, 7:3, and 8:2 significantly decreased the expression level of *COX2* in primary chondrocytes and RAW 264.7, respectively (Appendix A). Comparison of *COX2* expression between the single extracts and PV and GL mixture at 150 µg/mL showed that the inhibitory effect of the optimal extract was a PV and GL combination at an 8:2 ratio (Figure 1A,B) without cytotoxic effects at 50, 100, and 150 µg/mL (Figure 1C,D). Consequently, we used a PV and GL mixture at an 8:2 ratio (PG) for subsequent examination.

### 3.2. PG Decreases Inflammation in Mouse Primary Chondrocytes and RAW 264.7 Cells

To investigate the effect of PG extract on inflammation in RAW 264.7 cells and primary chondrocytes, catabolic and inflammatory-associated factors, such as *COX2*, *Mmp3,* and *Mmp13,* were measured by qRT-PCR. Treatment with IL-1β in chondrocytes and LPS in RAW 264.7 cells increased the mRNA expression levels of *COX2, Mmp3*, *Mmp9,* and *Mmp13*. However, the PG extract significantly reduced the mRNA expression of *COX2, Mmp3, Mmp9,* and *Mmp13* in both RAW 264.7 cells and primary chondrocytes (Figure 2A,B). Furthermore, Western blot analysis showed that protein levels of NF-κB and COX2 were decreased, and NO production was significantly decreased in RAW 264.7 after the PG extract treatment (Figure 3). These results suggest that the PG extract decreased inflammation in mouse primary chondrocytes and RAW 264.7 cells.

### 3.3. PG Inhibits Secretion of Prostaglandin E2 (PGE2) and Collagenase in IL-1β-Induced Primary Chondrocyte Inflammation

To further confirm the anti-inflammatory effect of PG extract on destructive mediators, the levels of secreted PGE2 and collagenase were investigated in IL-1β-induced primary chondrocytes. The administration of IL-1β and co-treatment with PG extract (100 and 150 µg/mL)induced inflammation of primary cultured chondrocytes. To measure the release of PGE2 and collagenase, the culture supernatant was concentrated using Vivaspin^®^, and the levels of destructive mediators were analyzed using an enzyme-linked immunosorbent assay (ELISA) kit. IL-1β-induced primary chondrocytes exhibited increased PGE2 and collagenase levels. However, treatment with the PG extract significantly suppressed the secretion of PGE2 and collagenase in IL-1β-induced chondrocytes. (Figure 4A,B). These results indicate that PG treatment inhibited the secretion of PGE2 and collagenase by downregulating the COX2, Mmp3, Mmp9, and Mmp13 expressions.

### 3.4. PG Prevents Subchondral Sclerosis in the DMM-Induced OA Mouse Model

Based on the in vitro results, the anti-inflammatory effects of PG extract were evaluated in an osteoarthritic mouse model by surgical DMM in vivo. DMM-induced mice were fed either food pellets or pellets containing different concentrations of PG extract (50, 100, and 200 mg/kg/d) for eight weeks. The left knee joint was removed and scanned on the last administration day using micro-computed tomography (micro-CT). Micro-CT images showed that DMM mice exhibited subchondral sclerosis and increased BMD in the subchondral region. However, treatment with the PG extract reduced DMM-induced sclerosis (Figure 5A, red) and increased BMD (Figure 5B). These results suggest that PG prevents tibial subchondral osteophyte formation in a mouse model of DMM-induced OA.

### 3.5. PG Reduces Cartilage Destruction in DMM-Induced OA Mouse Model

Histological analysis of articular cartilage destruction was performed to confirm the protective effects of PG extract on cartilage destruction in the DMM-induced OA model. By safranin O staining, DMM mice presented a loss of cartilage in the knee joint compared with the sham-operated group. In addition, immunohistochemical analysis showed that the DMM group exhibited upregulated expression of catabolic factors such as COX2, Mmp3, and Mmp13 in the articular cartilage. However, PG extract treatment significantly decreased cartilage destruction (Figure 6A, left) with reduced expression levels of catabolic factors (COX2, Mmp3, and Mmp13) compared to the DMM-induced OA group (Figure 6A, right). Administration of PG extract decreased DMM-induced vertical clefts and erosions extending to the calcified layer of the articular surface using the OARSI grade and quantified by the relative expression of COX2, Mmp3, and Mmp13 (Figure 6B).

### 3.6. PG Decreases Inflammatory Cytokines in the DMM-Induced OA Mouse Model

To examine the effects of PG on inflammatory cytokines, the secretion of IFN-γ, IL-1β, IL-6, and TNF-α was analyzed in a mouse model of DMM-induced OA. As expected, DMM mice exhibited elevated plasma levels of inflammatory cytokines IFN-γ, IL-1β, IL-6, and TNF-α. However, the PG extract significantly inhibited the plasma levels of OA-induced cytokines (Figure 7). Taken together, these results suggest that PG treatment prevents subchondral osteophyte formation by inhibiting catabolic factors and inflammatory cytokines in DMM-induced OA mice.

## 4. Discussion

Osteoarthritis (OA) is a common degenerative disorder, among which knee arthritis is the most common global disease [37]. Although many pharmaceutical drugs have been used for osteoarthritis treatment [38], their long-term side effects have been reported in older, frail, and comorbid patients [39]. This study describes the potential effected of anti-inflammatory herbal plants, PV and GL, in vitro and in vivo.

In OA patients, chondrocytes are stimulated by external stress, leading to the upregulation of various catabolic molecules such as IL-1, COX2, and MMPs [40]. COX2 expression is upregulated by inflammatory cytokines, promoting increased production of MMP3 and MMP13, which induces proteoglycan and collagen degradation with apoptosis of chondrocytes [41]. In addition, increased inflammatory responses in osteoarthritis upregulate the release of destructive mediators such as NO, prostaglandin, and collagenase [42]. Prostaglandin, synthesized by the cyclooxygenase (COX) enzyme, is a derivative lipid molecule that regulates homeostasis and inflammation in the body [43]. It has been reported that osteoarthritic patients present elevated levels of PGE2 and collagenase in articular cartilage, promoting the degradation of collagen in cartilage [44,45]. Hence, NO, PGE2, and collagenase play important roles in regulating knee remodeling during inflammation. Studies have reported that both PV and GL promote anti-inflammatory activities [27,46]. In this study, the PG combination showed synergistic effects on inflammation in primary chondrocytes and RAW 264.7 cells without cytotoxic effects by reducing catabolic factors. In addition, treatment with the PG extract decreased inflammation-mediated secretion of PGE2, collagenase, and NO production. The results showed that upregulation of NF-κB, COX2, and MMPs induced by external stress inhibited using PG treatment prevented the secretion of inflammatory factors. This study suggests that the PG extract has protective effects against inflammation in primary chondrocytes and RAW 264.7 cells.

The knee joint comprises several compartments, including articular cartilage, the meniscus, and ligaments [47]. Articular cartilage is a white connective tissue that provides a smooth, lubricated surface for the articulation to protect the ends of bones where they come together to form joints [48]. The meniscus of the knee joint is a crescent-shaped wedge of fibrocartilage located between the femoral condyle and tibial plateau, which provides increased stability to femorotibial articulation, distributes axial load, absorbs mechanical stress, and provides lubrication to the knee joint [49]. Various types of osteoarthritic mouse models, including surgical, chemical, and spontaneous methods, have been reported, and the DMM model has been widely used to evaluate degenerative OA [50]. Patients with OA typically exhibit high levels of BMD in physically damaged knees [51] and an osteosclerotic phenotype in the subchondral region of the tibial plateau [52]. In addition, a sclerotic phenotype of the OA model was observed in DMM-induced mice [53]. Similarly, our results showed that the OA mouse model induced by DMM showed subchondral osteophyte formation with increased BMD. However, micro-CT images of the left knee joint showed that treatment with PG extract inhibited the DMM-induced BMD increase in the tibial plateau. These results suggest that PG prevents subchondral sclerosis in a mouse model of DMM-induced OA.

High levels of inflammatory cytokines, including IFN-γ, IL-1β, IL-6, and TNF-α, have been detected in OA patients, promoting the activation of MMPs and catabolic factor synthesis [54]. These processes are present in damaged knees with a reduction in type 2 collagen and proteoglycan expression and destruction of articular cartilage [55]. As histological analysis of destructive knees is important for degenerative arthritis studies [56], cartilage destruction and inflammatory responses were measured by detecting proteoglycan content using safranin O staining and immunohistochemistry of catabolic factors (COX2, Mmp3, and Mmp13) in the articular cartilage. Our results showed that the PG extract reduced articular cartilage destruction and decreased the inflammatory cytokine levels of IFN-γ, IL-1β, IL-6, and TNF-α and catabolic factors in DMM-induced OA mice. These results suggest that PG treatment inhibits DMM-induced destruction and inflammation with osteophyte formation in the subchondral cartilage region by reducing inflammatory responses and cytokine production (Figure 8). 

Previous studies have reported that phytopharmaceutical compounds identified from natural products could exert beneficial effects on patients with OA [57,58]. However, a limitation of our study is that it is unclear whether PG presents clinical efficacy in the treatment of patients with knee osteoarthritis. In addition, this study investigated only male DMM mice; however, it is uncertain whether the anti-osteoarthritic effects of PG are sex specific. Our in vitro and in vivo results showed beneficial effects in OA experimental models, and it is necessary to study the application of PG extract to humans in future studies. Consequently, PG may be a potential candidate for the inhibition and prevention of osteoarthritis.

## 5. Conclusions

In summary, this study investigated the effects of PG extracts in vitro and in vivo. The PG extract decreased the expression of catabolic factors and inflammatory mediator levels in primary chondrocytes and RAW 264.7 cells. Administration of the PG extract in a DMM-induced OA mouse model protected articular cartilage destruction by inhibiting inflammatory processes. These results suggest that the PG extract can be used as an alternative herbal medicine to treat patients with OA.

## Figures and Tables

**Figure 1 antioxidants-12-00047-f001:**
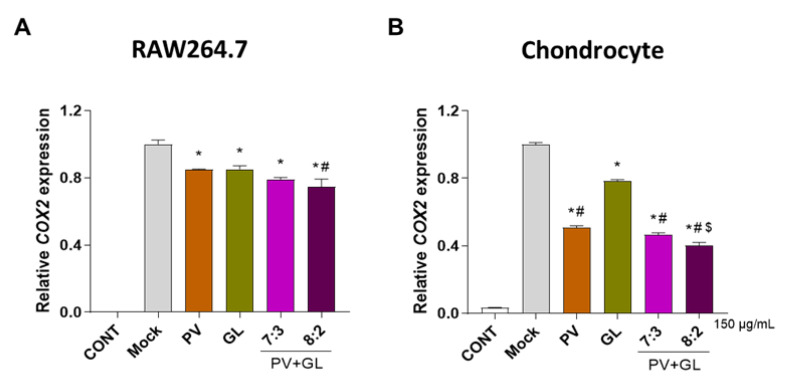
**Effects of PV and GL extract on Raw 264.7 cells and primary chondrocytes.** RAW 264.7 cells were incubated individually with PV and GL extract and their combined mixture (PG extract) with different ratios (7:3 and 8:2) for 24 h. Primary chondrocytes were incubated individually with PV and GL extract and their combined mixture (PG extract) with different ratios (7:3 and 8:2) for 48 h. (**A**,**B**) Relative *COX2* expression was determined by qRT-PCR. (**C**,**D**) Cell viability was examined by WST assay. * *p* < 0.05 vs. Mock, ^#^
*p* < 0.05 vs. GL, ^$^
*p* < 0.05 vs. PV. CONT; non-treated group, PV; *Prunella vulgaris* extract, GL; *Gentiana lutea* extract.

**Figure 2 antioxidants-12-00047-f002:**
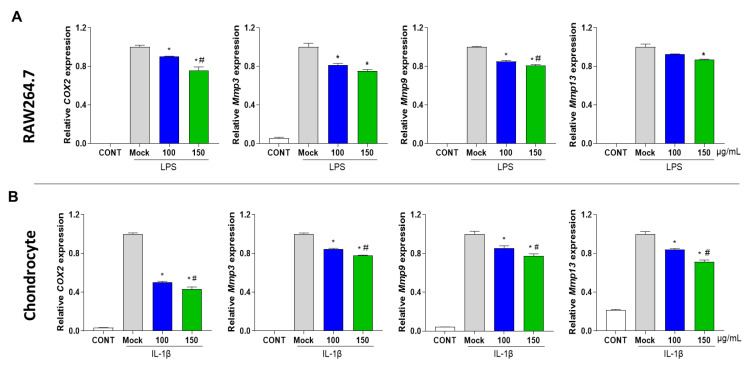
**Effect of PG extract on inflammatory-associated factors in RAW 264.7 and primary chondrocytes.** (**A**) RAW 264.7 cells were co-incubated with LPS and different concentrations (100 and 150 μg/mL) of PG extract (8:2 ratio) for 24 h. (**B**) Primary chondrocytes were co-incubated with IL-1β and different concentrations (100 and 150 μg/mL) of PG extract (8:2 ratio) for 48 h. Relative expression levels of *COX2*, *Mmp3*, *Mmp9,* and *Mmp13* were determined by qRT-PCR. * *p* < 0.05 vs. Mock, ^#^
*p* < 0.05 vs. 100. CONT; non-treated group.

**Figure 3 antioxidants-12-00047-f003:**
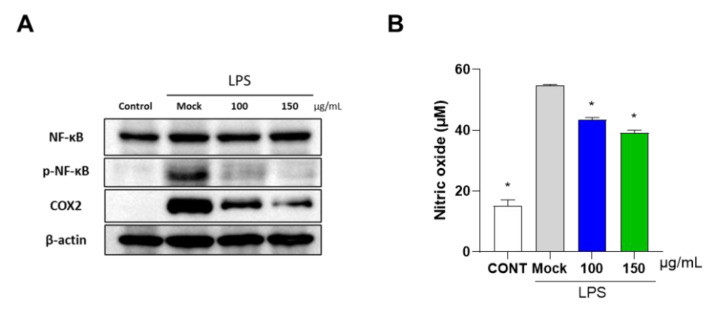
**Effect of PG extract on inflammatory signaling cascade of OA in RAW 264.7 cells.** RAW 264.7 cells were co-incubated with LPS and different concentrations (100 and 150 μg/mL) of PG extract (8:2 ratio) for 24 h. (**A**) Protein levels of COX2, NF-κB and p-NF- κB were subjected to Western blot analysis. β-actin was used as a loading control. (**B**) NO production was examined using nitrate/nitrite assay. * *p* < 0.05 vs. Mock, CONT; non-treated group.

**Figure 4 antioxidants-12-00047-f004:**
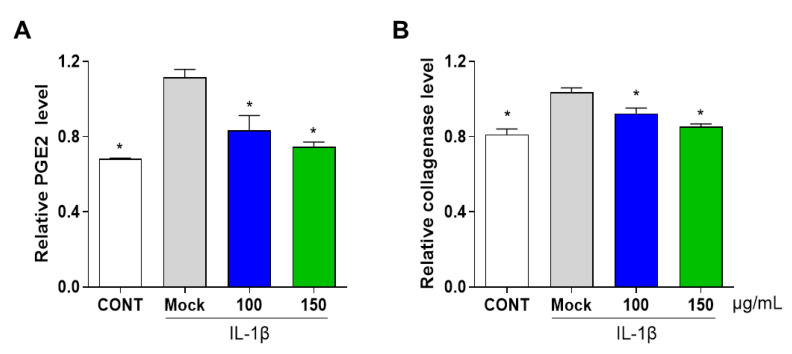
**Effect of PG extract on destructive mediators in IL-1β induced primary chondrocyte inflammation.** Primary cultured mouse chondrocytes were incubated with different concentrations of PG extract (100 and 150 μg/mL) in the presence of IL-1β. (**A**) PGE2 and (**B**) collagenase levels in the cell supernatants were examined. * *p* < 0.05 vs. Mock. CONT; non-treated group.

**Figure 5 antioxidants-12-00047-f005:**
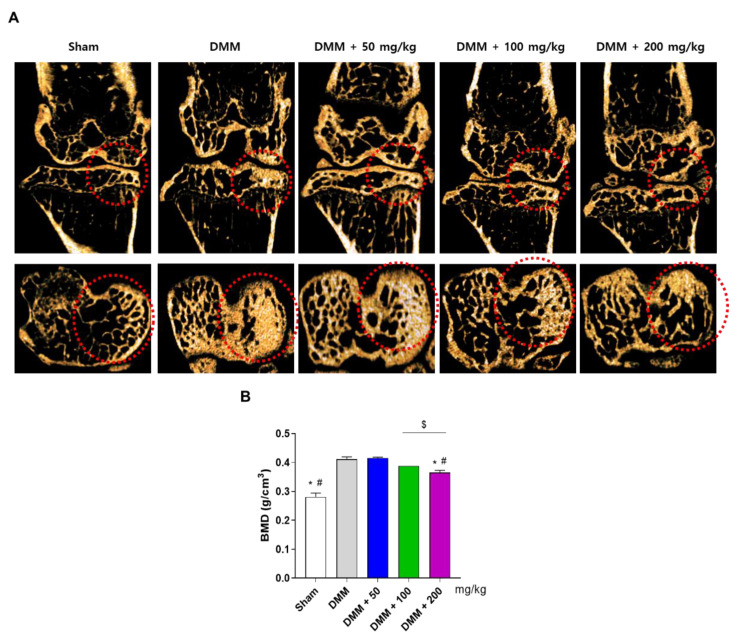
**Effects of PG extract on DMM-induced mouse OA model.** Mice were induced OA by DMM, and different concentrations of PG extract (50, 100, and 200 mg/kg) were administered for 8 weeks. (**A**) Subchondral knee joint was scanned by micro-CT. (**B**) BMD of the subchondral knee joint was determined. * *p* < 0.05 vs. DMM, ^#^
*p* < 0.05 vs. DMM + 50, ^$^
*p* < 0.05 vs. Sham. Sham; non-treated group.

**Figure 6 antioxidants-12-00047-f006:**
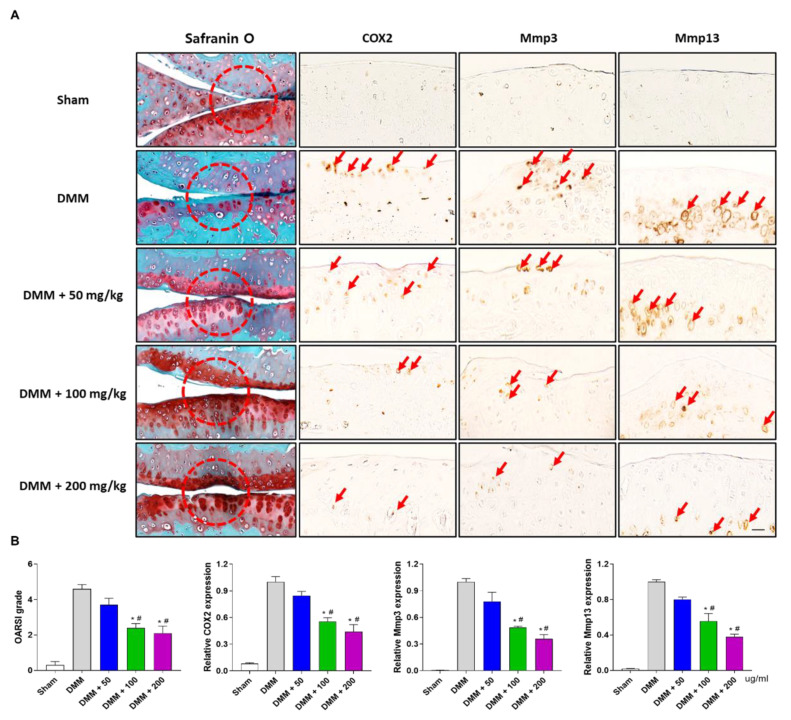
**Effects of PG extract on catabolic factors in a DMM-induced OA mouse model.** (**A**) Representative images for safranin O staining and immunohistochemistry (IHC) for COX2, Mmp3 and Mmp13 of articular cartilage. (**B**) OARSI scoring and quantification of IHC for COX2, Mmp3 and Mmp13 was determined. * *p* < 0.05 vs. DMM, ^#^
*p* < 0.05 vs. DMM + 50. Sham; non-treated group.

**Figure 7 antioxidants-12-00047-f007:**
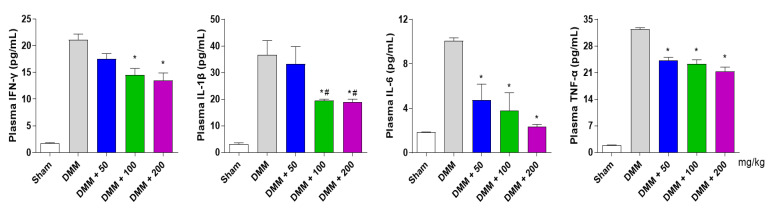
**Effect of PG extract on plasma levels of inflammatory cytokines in a DMM-induced OA mouse model.** Plasma levels of inflammatory cytokines, such as IFN-γ, IL-1β, IL-6, and TNF-α, were determined using cytokine/chemokine magnetic bead panel immunology multiplex assay. * *p* < 0.05 vs. DMM, ^#^
*p* < 0.05 vs. DMM + 50. Sham; non-treated group.

**Figure 8 antioxidants-12-00047-f008:**
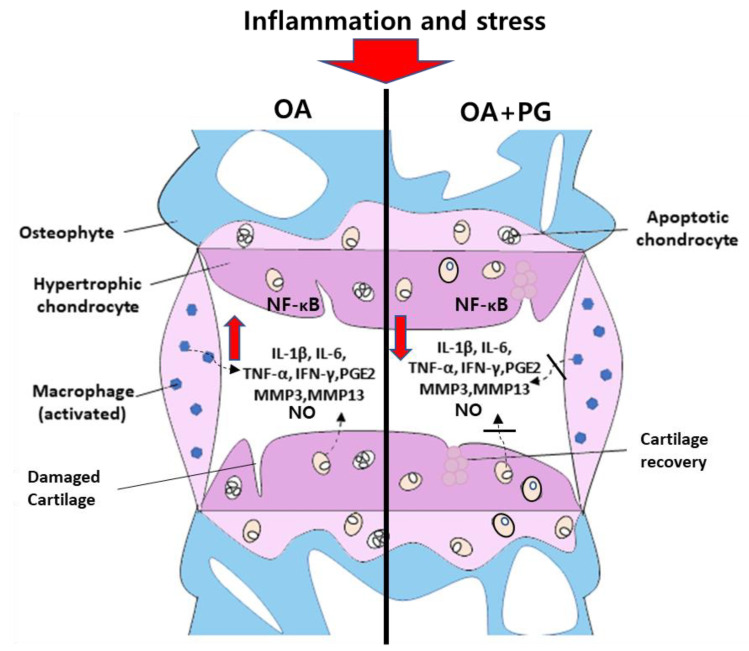
**Molecular mechanism for the inhibition of inflammatory cytokines using a PG extract**.

## Data Availability

The data presented in this study are available on request from the corresponding author.

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
