# Peer review of "Anti-Osteoarthritic Effects of Prunella Vulgaris and Gentiana Lutea In Vitro and In Vivo"

_antioxidants, 2022, doi:10.3390/antiox12010047_

Round 1

Reviewer 1 Report

Title: Anti-osteoarthritic effects of Prunella vulgaris and Gentiana lutea in vitro
and in vivo
 Authors: Seung-Hee Yun, Chang-Gun Lee, Seokjin Hwang, Jeonghyun Kim,
Laxmi
Prasad Uprety, Kang-Il Oh, Shivani Singh, Jisu Yoo, Hyesoo Jeong,
Yoonjoong Yong, Subin Yeo, Eunkuk Park, Seon-Yong Jeong

Comments:

The authors show an anti-inflammatory and anti-osteoarthritic effect of the combination of two phytochemicals both in a cell model and in a mouse model. Some aspects should be added to the content of the paper. Overall, I have the following suggestions for optimization:

Introduction:

1: Lines 37-44 need to be revised. Osteoarthritis does not only affect the knee joint. If the authors want to refer exclusively to knee osteoarthritis, this should be made clear in the definition and the mention of the number of patients.

2: Line 53: NF-kB should also be written out here. Its importance as an important transcription factor for inflammatory processes should be described. Its axis to Sox9 and the Src/PI-3K/AKT pathway in chondrocytes should also be mentioned,

Please add the following references:

doi: 10.3390/ijms22147645;

doi: 10.1371/journal.pone.0028663.

3: Lines 64-68: Phytochemicals that have already been successfully researched could be cited here as examples, e.g. curcumin (Please add the following references:

doi: 10.3390/nu14010041

doi: 10.3390/ijms22147645

and resveratrol: please add the following references:

doi: 10.1097/GME.0000000000000861;

doi: 10.1007/s12263-010-0179-5.

4: Lines 70-73: Please add why the authors specifically selected and combined PV and GL. The background is not obvious to the reader.

Figure 5: The images should be displayed slightly larger. Also, important structures could be marked and described in the text for the reader.

Discussion:

5: More should be written about your own results.

6: The limits of transferability of the present results to humans should be discussed.

7: If clinical trials are not yet available for PV and GL, the usefulness could be supported with successes of other phytopharmaceuticals in clinical trials. Please add the following references:

doi: 10.1002/ptr.6907;

doi: 10.1186/s12906-020-03105-0.

Formatting corrections should be made throughout the text:

8: line 90/91: punctuation.

9: Line 202: Figure 1A and B.

10 Page 7/8: Numbering of chapters.

12: Explain abbreviations only when first mentioned (e.g., COX) and use abbreviations consistently (e.g., PGE2, MMP).

Author Response

Best regards,

Reviewer 2 Report

The authors of this article demonstrated the anti-osteoarthritic effects of Prunella vulgaris (PV) and Gentiana lutea (GL) on primary chondrocytes and macrophages (RAW 264.7) plus an in vivo DMM mouse model. In vitro data on chondrocytes and macrophages demonstrate a significant reduction in COX2, MMP3 and MMP13 expression, whilst these were results were replicated in the DMM model alongside reduced osteophyte formation and improvements in OARSI grade.

The results presented are interesting and provide an alternative solution for the treatment of osteoarthritis. The authors should consider the following,

1.    What was the rationale to use mouse chondrocytes and macrophages rather than human ? I understand with respect to the animal model, although human data would be of interest.

2.       Why were the herbal compounds provided in food supplement rather than direct injection into the joint ? It is well-known that mouse cartilage is able to regenerate and so clarification on this rationale is required.

3.  Do the authors have data o to understand the mechanism for the inflammatory cytokine inhibition via PG application ? Western blots for NF-kB should be presented, along with a schematic figure.

4.   Can the authors present data on reactive oxygen species and nitric oxide to show that other inflammatory molecules are also reduced ? Nitric oxide is a well-known catabolic factor in OA and the effect on PG on this molecule should be presented.

5.  How would these herbal extracts be applied in a clinical situation to patients ? Authors should add a comment in the discussion on this part.

Author Response

Best regards,

Round 2

Reviewer 1 Report

The authors answered all questions to my satisfaction. No further questions.

Reviewer 2 Report

The authors have addressed my concerns appropriately.